# Streamlining the Municipal Waste Management System in the City of Nitra (Slovak Republic) Based on a Public Survey

Zuzana Pucherová [1] , Imrich Jakab [1,*], Anna Báreková [2] and Jarmila Králová [3]

1 Department of Ecology and Environmental Studies, Faculty of Natural Sciences,
   Constantine the Philosopher University, 949 01 Nitra, Slovakia; zpucherova@ukf.sk
2 Institute of Landscape Engineering, Faculty of Horticulture and Landscape Engineering,
   Slovak University of Agriculture, 949 76 Nitra, Slovakia; anna.barekova@uniag.sk
3 Arboretum Mlyňany Slovak Academy of Science, 951 52 Vieska nad Zitavou, Slovakia;
   Jarmila.Kralova@savba.sk
* Correspondence: ijakab@ukf.sk

**Abstract:** The main problems of the city of Nitra (Slovak Republic) in the field of municipal waste management include: 1. High production of municipal waste per capita; 2. Low rate of its separation; 3. High landfill rate; 4. No opportunity for composting; 5. Establishment of illegal landfills in the city; 6. Low waste prevention rate. To identify the attitudes and opinions of the respondents, and to evaluate certain behavioural practices of the inhabitants of Nitra in the management of municipal solid waste, we used a structured questionnaire (realized in 2020). The results of the questionnaire correspond to the behaviour of 4911 inhabitants of the city (6.46%). This paper evaluates the respondents' answers, which could be utilised by the local government—not only for a more appropriate setting of municipal waste management and separation, but also in waste prevention and monitoring changes in the consumer behaviour of city residents. The degree of separation in individual housing construction (IHC) and complex housing construction (CHC) was statistically evaluated and compared separately. For paper and glass, a higher degree of separation was reflected in CHC; while conversely, households living in CHC avoided bio-waste and kitchen waste more than IHC households. The most common reason for not participating in the separate collection was the lack of collection containers, the distance of containers from their households, or the low frequency of their collection. The results of the questionnaire show the need for more rigorous education about waste generation, its proper separation, and its prevention.

**Keywords:** municipal solid waste; separate waste collection; waste management; Nitra city; questionnaire survey

## 1. Introduction

Consumerism in our daily lives is the main reason for the huge production of waste, and people are now very quickly succumbing to the charm of consumerism, in which even durable products are slowly becoming consumer products. Several environmental problems come hand in hand with the increasing production of waste.

As the world hurtles toward its urban future, the amount of municipal solid waste (MSW), one of the most important by-products of an urban lifestyle, is growing even faster than the rate of urbanization [1].

Excessive production of municipal waste and incorrect or missing methods of waste separation in households are currently among the main environmental problems of most municipalities not only in the Slovak Republic (SR) but also abroad. It is the local governments, in cooperation with the local population, that play a major role in the production of municipal waste as part of the waste management. For example many urban areas are facing severe problems in managing 10 to 50 metric tonnes of waste per day in Sri Lanka [2], the urban centres in Bangladesh together generate about 23,688 tonnes/day of

MSW [3], 1120 tonnes of waste per day in Depok, Indonesia [4] or 13,327 tonnes/day in 2017 in Bangkok, Thailand [5], in Tehran, the capital and the most urbanized city of Iran with 8,693,706 inhabitants was produced 1000–1200 kg/capita/day of MSW during the COVID-19 pandemic [6], etc. Pursuant to Directive 1999/31/EC on the landfills of waste par. 3 [7], it is necessary to promote waste prevention, recycling, and recovery of waste, as well as the use of secondary raw materials and energy obtained from waste treatment, to protect natural resources and avoid uneconomical land use.

MSW consists of various substances and decomposes very slowly in landfills. MSW is a heterogeneous waste and composition of the waste varied from place to place [8]. MSW is the natural result of human activities, and its generation modelling is of prime importance in designing and programming management system with it [9]. Every year, the amount of waste produced at the European and global level increases. Almost 225 million tonnes of municipal waste were generated in the EU in 2019. This corresponds to 502 kg per person [10]. The quantity of municipal waste varies according both to change in season of the year and to a variety of impact factors, such as: socioeconomic status of household, demography, or environmental awareness [9,11,12]. In 2019, each citizen in the SR produced 435 kg of MSW (approximately 1.2 kg per day) and in the last 10 years the production of municipal waste in the SR increased by 35% [13]. Although slightly more waste is being generated, the total amount of municipal waste landfilled in the European Union (EU) has more than halved since 1995: from 121 million tonnes (286 kg per person) in 1995 to 54 million tonnes (120 kg per person) in 2019 [10]. Landfilling remains the most common method of waste management in the SR (2019-52%), making the SR one of the EU countries with the highest share in this field.

In accordance with the strategic document in the field of waste management in the SR for the years 2021–2025 [14], the main goal of waste management in the SR until 2025 is to divert waste away from landfilling. Binding objectives and measures for MSW about their landfilling and disposal are set in the SR as follows:

- Increase the rate of separate collection of MSW, including its preparation for re-use and recycling to 55% by 2025, 60% by 2030, and 65% by 2035,
- Decrease landfilling of mixed MSW to 25% by 2030 and to 10% by 2035,
- Reduce the share of biodegradable municipal waste in MSW to 25% by 2025,
- An obligation to recycle at least 65% of all packaging waste by weight by 2025 and 70% by 2030.

The municipality of Nitra is fully aware of the setting of objectives in the field of waste management for the years 2025, 2030 and 2035. Therefore, to reduce the production of MSW and improve the current unsatisfactory management, it decided to involve the city residents in solving this problem. Through the questionnaire, residents were able to express their views and attitudes for the first time, which could potentially be used in the gradual implementation of measures related to the six current main problems of the city. These were defined by the local government in cooperation with the members of the Waste working group, whose members are representatives of the local government (Nitra City Hall), members of the Commission for Environment, Public Order and Municipal Activities, representatives of the Nitra municipal services, Ltd., experts from 2 universities operating in the territory (Constantine the Philosopher University in Nitra and Slovak University of Agriculture in Nitra) and experts from the third sector (Friends of the Earth Association).

1. **High production of mixed MSW per capita**

In 2010–2020, every inhabitant of the city of Nitra produced an average of 317.53 kg/year of mixed MSW. In 2010, 28,046.57 tonnes of mixed MSW were produced and in 2020 this amount decreased by 4228.89 tonnes (Table 1), while the number of inhabitants in the city decreased by 7416 during this period.

**Table 1.** Amount of mixed municipal waste produced, separated collection, separation rate and quantities per capita of the city of Nitra in the years 2010–2020.

| Year | Quantity in Tonnes | | | | | | | | | | |
|---|---|---|---|---|---|---|---|---|---|---|---|
| | 2010 | 2011 | 2012 | 2013 | 2014 | 2015 | 2016 | 2017 | 2018 | 2019 | 2020 |
| mixed municipal waste | 28,046.57 | 25,702.09 | 24,604.21 | 24,797.63 | 24,872.93 | 24,107.76 | 24,595.07 | 24,160.36 | 24,175.10 | 23,813.18 | 23,817.68 |
| separated collection | 3482.42 | 3187.72 | 2845.16 | 2898.23 | 3447.69 | 3526.66 | 3870.27 | 3910.17 | 4021.45 | 4499.08 | 4658.31 |
| total | 31,528.99 | 28,889.81 | 27,449.37 | 27,695.86 | 28,320.62 | 27,634.42 | 28,465.34 | 28,070.53 | 28,196.55 | 28,312.26 | 28,475.99 |
| separation rate | 11.00% | 11.00% | 10.40% | 10.50% | 12.20% | 12.80% | 13.60% | 13.90% | 14.30% | 15.90% | 16.40% |
| population | 83,444 | 78,875 | 78,607 | 78,351 | 78,033 | 77,670 | 77,374 | 77,048 | 76,655 | 76,533 | 76,028 |
| quantity of municipal waste per capita (kg) | 336.11 | 325.86 | 313.00 | 316.49 | 319.75 | 310.39 | 317.87 | 313.56 | 315.38 | 311.15 | 313.28 |
| quantity of separa-ted waste per capita (kg) | 41.73 | 40.42 | 36.20 | 36.99 | 44.18 | 45.41 | 50.02 | 50.75 | 52.46 | 58.79 | 61.27 |

Source: the Nitra municipal services, Ltd. & Statistical Office of the SR (http://datacube.statistics.sk; accessed on 12 September 2021).

2. **Relatively low rate of municipal waste separation**

In the years 2010–2020, each inhabitant of the city of Nitra separated an average of 47.11 kg/year of separated commodities (paper, plastics, glass, metals, multilayer composite materials, discarded electronic equipment, batteries, and accumulators). It was not until 2018 that a uniform methodology for calculating the separation rate was introduced in the territory of the SR, which was developed by the Ministry of the Environment of SR. The separation rate in the city of Nitra was 45.10% (2018) and we recorded a decrease at 42.20% (2019) and 40.80% (2020) (Table 1). In terms of this methodology, we calculated the rate of separation of municipal waste in the city of Nitra in previous years (2010–2017) (Table 1).

3. **High landfilling rate**

Although the production of mixed MSW and its subsequent landfilling has a gradually declining trend in the city of Nitra in the period from 2010 to 2020, mixed MSW is nevertheless disposed of only by the landfill method due to the absence of an incinerator. In 2010 a total of 28,046.57 tonnes of mixed MSW was transported to the regional landfill Tekovská ekologická, s.r.o. in Nový Tekov and in 2020 this amounted to 23,817.68 tonnes of mixed MSW.

4. **No opportunity for composting**

Since September 2015, a regular two-week collection of biodegradable waste (biowaste) from the gardens of family houses (IHC) has been introduced in the city of Nitra. These residents were not provided with their own composters, which could reduce the number of bio-waste is generated. This type of collection is not carried out within urban housing estates in CHC. In several places in CHC, the introduction of community composting is required, which we can currently meet only in the wider community of residents of the housing estate block at one location in the Chrenová housing estate.

5. **The creation of illegal landfills in the city**

In the city of Nitra, we have registered several illegal landfills in recent years. In the period from 2017 to 2020, according to official data of the municipal office within the city of Nitra, a total of 68 illegal landfills were removed (11 in 2017, 8 in 2018, 27 in 2019, and 22 in 2020). In 2019, 44 illegal landfills were mapped in the city of Nitra, with the largest number in the district of Nitra city Staré Mesto [15].

6. **Low waste prevention rate**

Based on data on the amount of municipal waste generated per capita, the rate of municipal waste separation and the quantity of various types of waste handed over by city residents at collection yards, the rate of waste prevention seems to us to be minimal.

2. **Materials and Methods**

*2.1. Study Area*

The city of Nitra (48.3121619N, 18.0881014E) is in the western part of the SR. It has an area of 100.48 km$^2$ and its territory is divided into 12 districts (Figure 1). Within the city there is individual housing construction (IHC) (especially in the parts of Zobor, Kynek, Párovské Háje, Mlynárce, Veľké Janíkovce, Dražovce, Čermáň, Krškany, etc.) and complex housing construction (CHC) in the form of panel housing estates (especially in the districts of city Nitra-Chrenová, Klokočina, Čermáň, Diely, etc.). Nitra is currently the fifth largest city in SR. It is a regional (Nitra region) and at the same time a district town (Nitra district). As of 31 December 2020, the city had a population of 76,028 inhabitants. The population density is 756.65 inhabitants per 1 km$^2$. Of the total area of the city of Nitra (10,047.87 ha), the largest area is formed by agricultural land (5590.80 ha, 55.64%, of which arable land covers 4689.89 ha), built-up areas and courtyards (1890.66 ha, 18.82%), forest land (1379.01 ha, 13.72%), other areas (1022.70 ha, 10.18%) and water areas (164.70 ha, 1.64%) [16]. The electrical, engineering, automotive, chemical and food industries have the most significant representation in the city of Nitra. The city has seen a great boom in

industrial production in recent years, especially with the construction and commissioning of the Jaguar Land Rover Nitra industrial plant (2018). The city of Nitra is interesting in terms of the occurrence of natural, cultural, and historical monuments and recreational facilities. There are 2 universities in Nitra: the Slovak University of Agriculture and the University of Constantine the Philosopher in Nitra. The network of schools is complemented by primary and secondary schools. At the same time, health care is provided in hospitals and polyclinics.

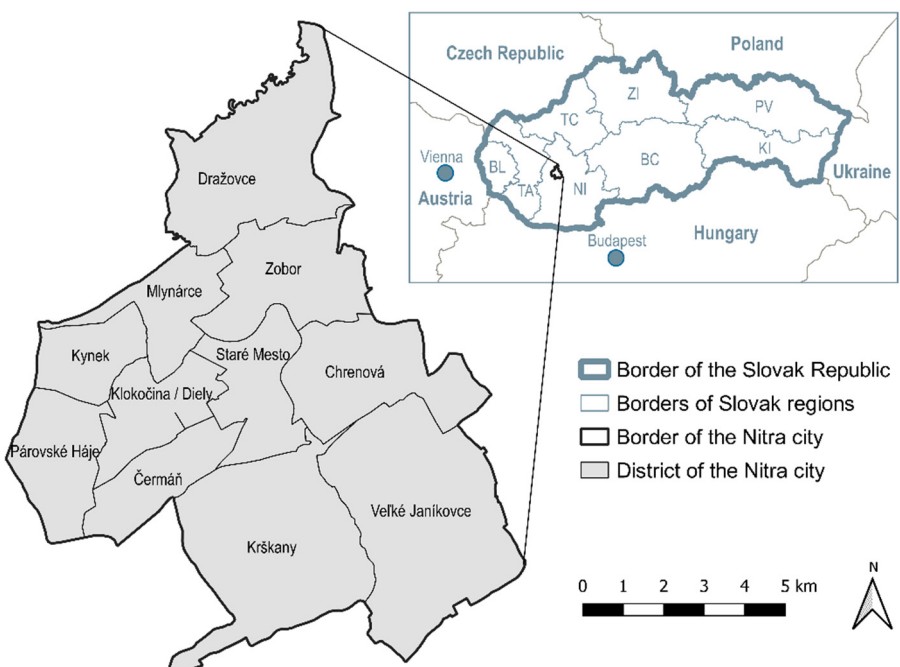

**Figure 1.** The localization of the study areas in the Slovak Republic.

The local government of the city of Nitra directs the activities and activities of the city in the field of protection and creation of the environment, including waste management and municipal waste management. The company the Nitra Municipal Services Ltd. collects and transports municipal, other and hazardous waste for the city of Nitra, as well as for some surrounding municipalities and industrial customers. The company has a revised integrated management system policy (ISO 9001, ISO 14001, and OHSAS 18001). For the citizens they provide collection of mixed and sorted municipal waste, while the collection of separated waste, separation, subsequent recovery, and recycling of these types of waste is financed by the ENVI-PAK Producer Responsibility Organization. In 2012, a municipal composting plant was put into operation, which has an annual processing capacity of approximately 16,000 tonnes of biodegradable waste. There are also 6 collection yards available for citizens in the city, where citizens can hand over e.g., construction waste, edible fats and oils, hazardous waste, discarded electrical and electronic equipment, bulky waste, batteries, accumulators, fluorescent lamps, but also all types of waste of separate collection (paper, plastics, metals, glass, bio-waste, multilayer composite materials) (Figure 2). Probably the most significant change occurred in the commodity type-"construction debris". Since 2016, collection yards have seen a reduction in the amount of small construction waste handed over, given that, in accordance with the generally binding regulations, the holder of such waste pays €0.05 for every kilogram. Until 2016, the small construction debris was accepted at the collection yards for free (Figure 2).

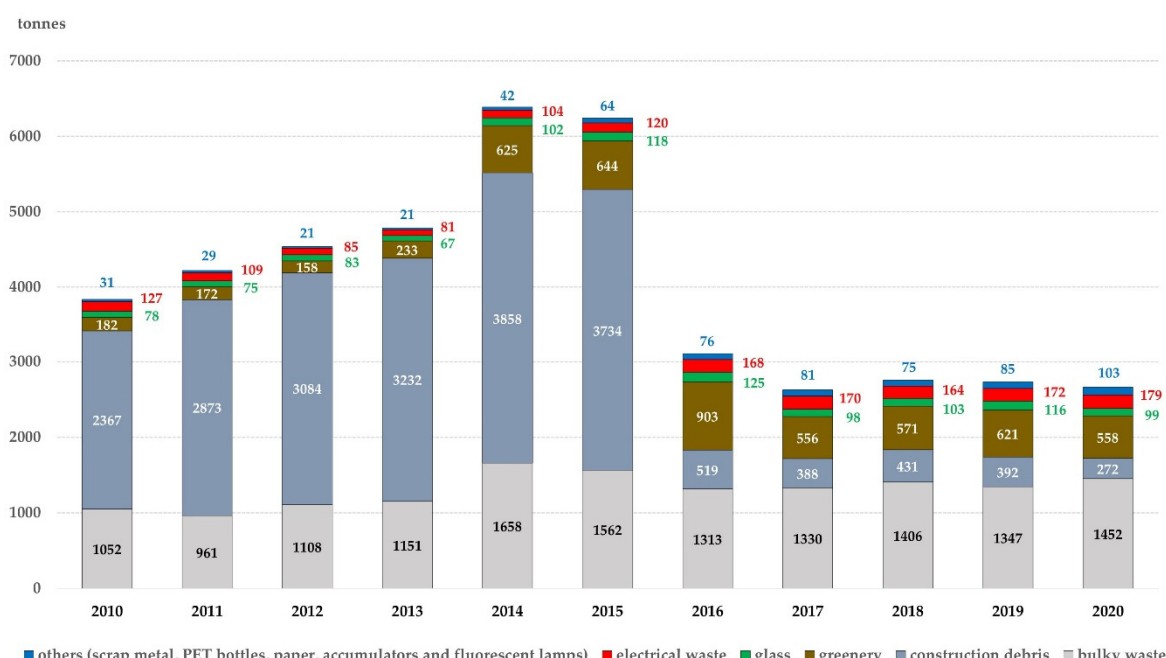

**Figure 2.** Types and quantity of waste handed over at collection yards within the territory of the city of Nitra in the years 2010–2020.

From 01.01.2010, the basic components (paper, plastics, metals, glass) must be separated in the cities and municipalities of the SR (Figure 3). Since September 2015, the city of Nitra has introduced a regular two-week collection of biodegradable waste from family home gardens (in IHC), which has caused a significant increase in collected and processed quantities of bio-waste since 2016. In 2017, the amount of biodegradable waste collected in Nitra increased significantly for several reasons. For instance, brown containers for bio-waste were distributed to all households in family houses, timber was cut down in preparation for the construction of a bicycle path and the Jaguar Land Rover industrial plant and there was a change in the management of the city's composting plant, etc. (Figure 4). The collection of multilayer composite materials started from 01.05.2017. In the years 2010–2017, multilayer combined materials were collected only at collection yards. In the period from 2010 to 2020, mixed MSW was transported to the regional landfill Tekovská ekologická, s.r.o. in Nový Tekov (Levice district, Nitra region).

*2.2. Data and Methodology*

When preparing the questionnaire, we proceeded methodically and systematically in accordance with the scheme (Figure 5). The reason for the implementation of the questionnaire was the entry conditions themselves. Based on the current unsatisfactory situation related to the production and management of municipal waste in the city of Nitra, there are six current main problems: 1. High production of municipal waste per capita; 2. Relatively low rate of municipal waste separation; 3. High landfill rate; 4. No opportunity for composting; 5. Establishment of illegal landfills in the city, 6. Low rate of waste prevention.

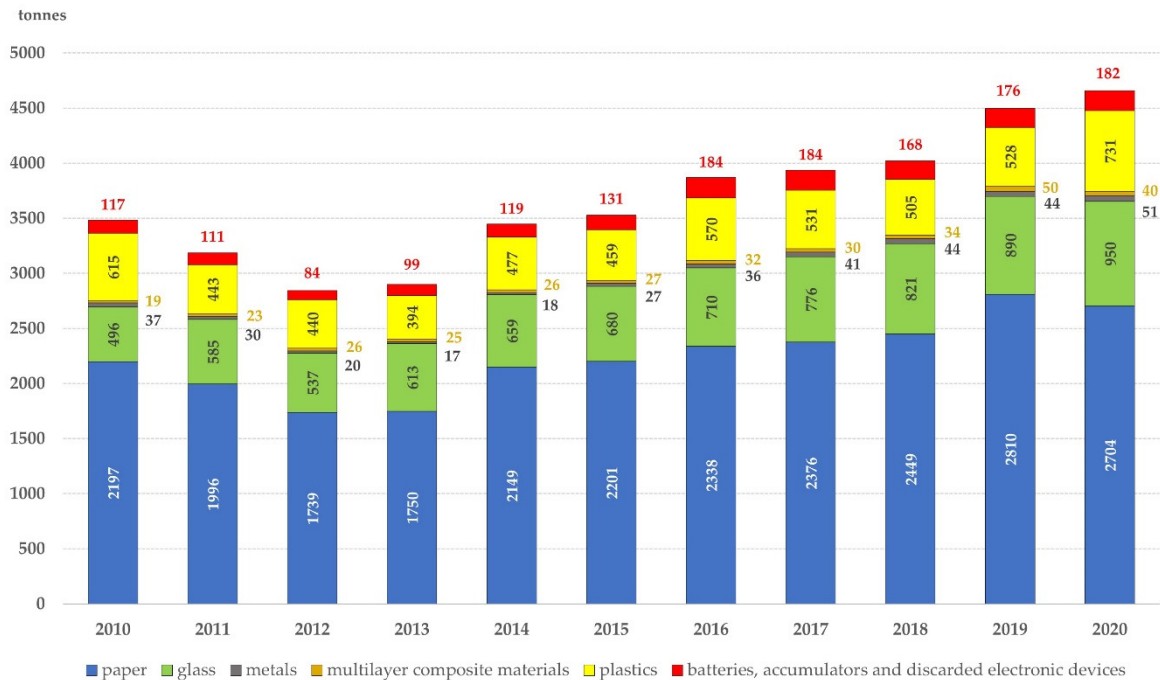

**Figure 3.** Quantities of separated commodities within the city of Nitra handed over for recycling in 2010–2020.

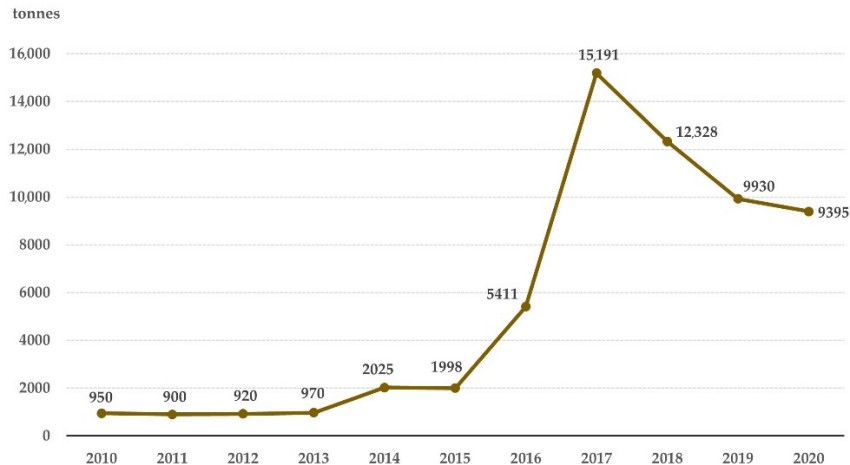

**Figure 4.** Quantities of collected and processed bio-waste within the city of Nitra in the years 2010–2020.

The implementation of waste management and the process of municipal waste management results from the valid legislation in the SR, especially from the Act of the National Council of the SR no. 79/2015 Coll. on waste, other applicable laws, decrees, regulations, and standards. Every year, the City Council approves the general binding regulation, which concerns the management of municipal waste and small construction debris in the city of Nitra. On the one hand, the municipality of Nitra has an annually approved city budget and certain material and technical equipment for the implementation of waste management, and on the other hand there is a lack of information, commitment and especially motivation of the population to responsible environmental behaviour. The city sets an annual flat fee for the collection of municipal waste, which is not in line with the "pay for what you throw away" principle. At the same time, we are gradually encountering examples of good practice, whether in the public sector, the private (business) sector, or in the so-called third (non-profit) sector.

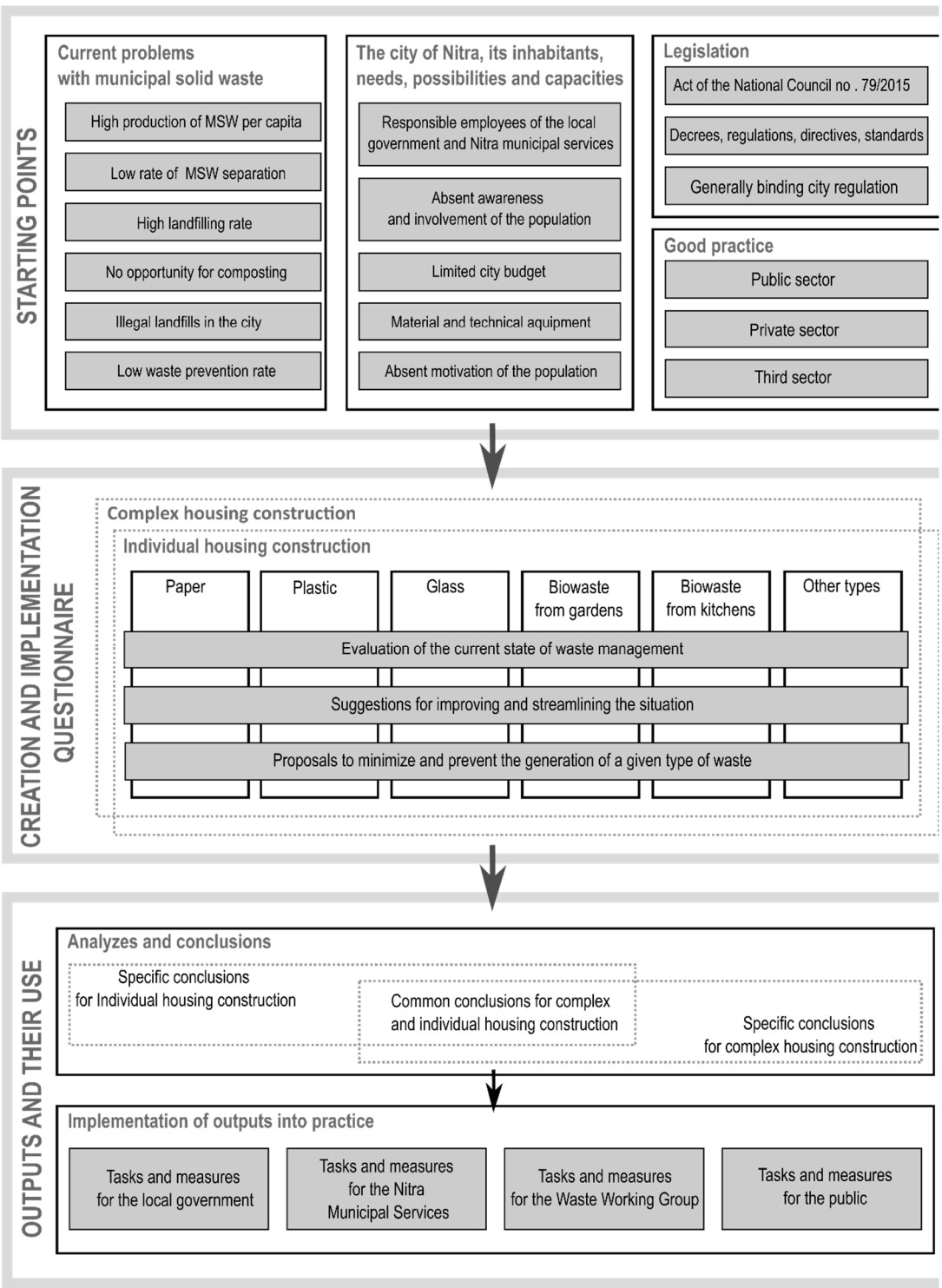

**Figure 5.** Scheme of the methodical procedure of questionnaire implementation.

The preparation and implementation of the questionnaire was provided by the Waste Working Group.

A structured questionnaire entitled Waste: A Survey of the Opinions of the Inhabitants of Nitra was realized in the period 14 May 2020–16 June 2020. A total of 23 questions were formulated through the freely available SurveyMonkey survey software (https://www.surveymonkey.com/r/oditray_Nitra; accessed on 12 May 2021).

The content of the questionnaire was divided into the following:

(a) Introductory questions (questions 1–3)—concerned general information about the respondents, i.e., the district of the city in which they live, in what type of housing unit they live, i.e., either a family house—individual housing construction (IHC) or an apartment in a residential building (complex housing construction (CHC)), and information about household members, specifying the number of adults and the number of children who use disposable nappies in the given household.

(b) Questions on individual types of waste—the basic areas of the questionnaire by type of waste: paper (questions 4–6), plastics (questions 7–9), glass (questions 10–12), bio-waste (question 13), kitchen waste (question 14) and cooking oil (question 15). For each type of waste, we were interested in the respondents' opinions regarding not only its separation, but also the current state of its collection and the reasons why the respondents do not separate the given type of waste. We were also interested in whether the respondents in family houses would be interested in composters that could potentially be provided by the city, or whether they would welcome the possibility of community composting (question 16). In question 17, we asked about other types of waste (e.g., metals, iron, polystyrene, furniture, post-expiry medicines, hazardous waste, small construction debris, etc.) that respondents separate, and which types of waste they hand over at collection yards (question 18).

(c) Questions about waste prevention issues—in two questions (questions 19 and 20), we were interested in finding out whether the respondents prevent the production of waste in some way and, if so, in what specific ways.

(d) Questions to spread information and education—a further two questions were aimed at the method of distribution of information and education in the field of waste management and waste prevention, and the way in which the respondents prefer to receive such information (questions 21 and 22).

(e) Other suggestions and comments—in the last question (question 23), we allowed the respondents to express, in the form of ideas, suggestions and comments, their views on how the current state of municipal waste management and separate collection in the city of Nitra could be improved.

For the first time, the inhabitants of the city of Nitra had the opportunity to express their opinion in the form of an anonymous questionnaire. In the past, the municipality did not use this method. The anonymous questionnaire was published in the period 14 May 2020 and 16 June 2020 through the freely available SurveyMonkey survey software (https://www.surveymonkey.com/r/oditrapy_Nitra; accessed on 12 May 2021).

Respondents chose from the options in six closed questions and 15 open questions; open questions allowed respondents to provide their own opinions, suggestions, and comments in the "other" category. In the specified period, 1495 respondents completed the questionnaire accessible to the public. We evaluate positively that residents from every district of the city of Nitra answered the questions of the questionnaire. Each respondent represented his or her entire household in the questionnaire (answering on behalf of the household members s/he stated in question 3). The results of the questionnaire thus correspond to the behaviour of 4911 inhabitants of the city (6.46%) of Nitra in the sorting of municipal waste. The questionnaire was acted upon by 858 respondents from CHC (57.39%) and 637 from ICH (42.61%). The number of household members was not given by three respondents, so we included them in the category "one household member", given that they commented on the other questions of the questionnaire. The number of inhabitants represented in the IHC and CHC households by the respondents in the questionnaire was approximately equal.

After completing the questionnaire survey, i.e., from 17 June 2020 (https://www.surveymonkey.com/results/SM-BWVZLH2F7/; accessed on 2 July 2021) the results of the questionnaire were analysed.

In the so-called open questions, i.e., no. 5, 6, 8, 9, 11–15, 17, 18, 20–23, respondents made suggestions for improvement or comments on the current situation. In some questions, respondents could provide more than one answer (questions 13, 14, 18, 20, 21, 22), so

the number of answers exceeds the number of respondents. In the case of these questions, we evaluated not only the percentage of the number of respondents, but also the total number of answers. For other questions, we evaluated the percentage of answers (options) only from the number of respondents who answered the question. In the case of question 23, we were provided with ideas, suggestions, and comments from respondents on how to improve the state of municipal waste management in the city of Nitra. During the analysis of the questionnaire, we drew specific conclusions typical for IHC, for CHC and conclusions common to IHC and CHC.

The dependency between housing (housing type: CHC and IHC) and separation of particular waste types was testing using Pearson´s contingency table chi-square test. A contingency table of standardised residuals was then used for mapping the relationship between the categories "separated" and "no separate".

In the final phase, based on the results of the questionnaire and the answers of the respondents, we set tasks and measures for: 1. The local government of the city of Nitra, 2. The Nitra Municipal Services, 3. The Waste Working Group and 4. The public.

Before preparing and implementing the questionnaire itself, we set out several hypotheses:

**Hypothesis 1 (H1).** *Housing (IHC, CHC) has an impact on the degree of sorting of individual types of waste.*

**Hypothesis 2 (H2).** *Respondents from IHC prefer container collection of separated municipal waste components over collection with bags.*

**Hypothesis 3 (H3).** *Respondents from CHC consider the number of containers for separated of municipal waste components to be insufficient.*

**Hypothesis 4 (H4).** *Respondents from IHC and CHC are not satisfied with the frequency of collection of separated components of municipal waste.*

**Hypothesis 5 (H5).** *Respondents from IHC and CHC are willing to separate the waste on condition that the collection containers are accessible and not very far away from the household.*

**Hypothesis 6 (H6).** *Most respondents consider it necessary to be more thoroughly informed about the classification of municipal waste.*

**Hypothesis 7 (H7).** *Respondents are not interested in home or community composting.*

**Hypothesis 8 (H8).** *Most respondents to the questionnaire do not try to limit the production of waste.*

The aim of our contribution was to obtain, analyze and interpret the attitudes and opinions of the inhabitants of the city of Nitra, through an anonymous questionnaire, and use them to streamline the municipal waste management system in the city. We approached IHC and CHC separately when assessing the results, which we also tested statistically, as it is necessary to evaluate these different housing units separately when setting up the waste management of the city of Nitra. The contribution can serve as a case study of purposeful cooperation in the field of waste management, in which representatives of the city of Nitra, along with waste management experts and citizens of the city of Nitra, all participated.

## 3. Results

### 3.1. Basic Information about Respondents

The territory of the city of Nitra is divided into 12 districts (Figure 1). In each of the districts of the city, there is also CHC and IHC. The largest number of inhabitants of the city live in the districts of Klokočina and Chrenová and, conversely, the smallest number of inhabitants live in the district of Mlynárce. Out of the total number of 1495 respondents, Klokočina (21.74%) and Chrenová (20.67%) were stated the most as the place of residence. These are also the largest housing estates in the city with a dominant housing development (flats in a residential building). Only in a smaller part of the housing estates of Chrenová (Chrenová II) and Klokočina (Klokočina III) there is no construction of family houses. Within the city of Nitra, the Zobor area has the largest number of households in the IHC (2169). This type of settlement however has a major position in other areas too, e.g., Dražovce, Krškany, Veľké Janíkovce, Kynek, Mlynárce and Párovské Háje. The smallest number of respondents came from the districts of Mlynárce and Dražovce. Of the total number of respondents (1495), 858 stated that they live in an apartment in an apartment building (57.39%) and 637 respondents stated that their type of housing unit is a family house (42.61%) (Table 2).

The most common number of household members, as stated in the respondents' answers, was four members (492 answers, 32.91%) (Table 3). In the next part, we wanted to know the number of children who use disposable nappies. These currently present a major problem in landfills of mixed municipal waste due to their weight, composition, and rate of decomposition.

In EU, it is estimated that absorbent hygiene products (baby nappies, feminine care, and incontinence products) represent between 1.5% and 6.3% of municipal solid waste [17]. The composition of disposable nappies is very diverse in terms of the materials used. Waste from nappies have a significant proportion of organic materials in their composition, but their destination in most of the countries is landfill or incineration [18]. Disposing of soiled nappies is a major global environmental problem as they constitute a large percentage of the municipal solid waste [19] and during the last several decades, disposable nappies have become a widely accepted alternative to cloth reusable nappies. It is a huge industry [20]. Used disposable nappies constitute a waste stream that has no established treatment method [21]. The disposal of used nappies is a critical eco-technological problem and their complete disposal by landfill takes almost 500 years [22]. A total of 1194 respondents (79.87%) commented on this part of the questionnaire. Of the number of respondents who expressed their opinion, 192 respondents (16.08%) answered by stating a certain number. From the answers we can state that the largest number of answers was one child (168), in eighteen cases it was two children, five respondents stated the number of children three and one respondent stated the number of four children. Based on the answers of the respondents at the time of filling in the questionnaire, a total of 223 children used disposable baby nappies.

**Table 2.** Number of households and population in the city of Nitra as of 31 December 2020 and inhabited parts of the city of Nitra by place of permanent residence of the respondents to the questionnaire.

| City District | CHC | | IHC | | The Total Population of the City District | Respondents from Districts of the City | | Population in Respondents' Households | |
|---|---|---|---|---|---|---|---|---|---|
| | Households | Population | Households | Population | | Total | CHC / IHC | Total | CHC / IHC |
| Staré Mesto | 5799 | 9542 | 592 | 2124 | 11,666 | 256 | 205 / 51 | 722 | 554 / 168 |
| Klokočina | 7388 | 15,982 | 638 | 2029 | 18,011 | 325 | 276 / 49 | 1005 | 796 / 209 |
| Diely | 3100 | 7405 | 58 | 192 | 10,505 | 80 | 75 / 5 | 243 | 223 / 20 |
| Kynek | 6 | 16 | 339 | 1092 | 1108 | 37 | 0 / 37 | 134 | 0 / 134 |
| Mlynárce | 13 | 26 | 173 | 544 | 570 | 21 | 4 / 17 | 76 | 10 / 66 |
| Párovské Háje | 39 | 108 | 324 | 1080 | 1188 | 38 | 5 / 33 | 126 | 14 / 112 |
| Veľké Janíkovce | 19 | 48 | 647 | 2084 | 2132 | 64 | 0 / 64 | 255 | 0 / 255 |
| Chrenová | 6983 | 12,387 | 769 | 2315 | 14,702 | 309 | 230 / 79 | 990 | 680 / 310 |
| Zobor | 420 | 926 | 2169 | 6479 | 7405 | 175 | 10 / 165 | 651 | 25 / 626 |
| Čermáň | 1619 | 2839 | 840 | 2568 | 5407 | 104 | 46 / 58 | 335 | 124 / 211 |
| Krškany | 187 | 538 | 782 | 2564 | 3102 | 55 | 6 / 49 | 200 | 15 / 185 |
| Dražovce | 19 | 27 | 526 | 1829 | 1856 | 31 | 1 / 30 | 174 | 3 / 171 |
| Total | 25,592 | 52,329 | 7857 | 24,900 | 74,744 | 1495 | 858 / 637 | 4911 | 2444 / 2467 |

Note: As of 31 December 2020, there were a total of 2485 citizens who have the stated address of permanent residence specifying only Nitra without the indication of a specific street. The total population of the city of Nitra was 77,229 as of the stated date. Source: Municipal office in Nitra, Explanations: CHC—complex housing construction, IHC—individual housing construction.

**Table 3.** Number of household members in respondents' answers in the city of Nitra.

| Household Members | 1 | 2 | 3 | 4 | 5 | 6 | 7 | 8 | 9 | 10 | 15 | 22 | 40 | Total |
|---|---|---|---|---|---|---|---|---|---|---|---|---|---|---|
| Number of respondents | 98 | 334 | 399 | 492 | 127 | 29 | 10 | 1 | 1 | 1 | 1 | 1 | 1 | 1495 |
| Percentage (%) | 6.55 | 22.34 | 26.68 | 32.91 | 8.49 | 1.94 | 0.67 | 0.07 | 0.07 | 0.07 | 0.07 | 0.07 | 0.07 | 100.00 |
| Total population in households | 98 | 668 | 1197 | 1968 | 635 | 174 | 70 | 8 | 9 | 10 | 15 | 22 | 40 | 4911 |

### 3.2. Waste Separation

#### 3.2.1. Paper

Question on whether the respondents separate the paper waste was answered by 1434 respondents (95.92%). The answer "yes" clearly dominated. Overall, 1254 respondents (87.45%) answered positively, of which 747 were respondents from CHC and 507 from IHC, and 180 respondents (12.55%) chose the "no" option, of which 86 respondents were from CHC and 94 from IHC.

We were interested in whether the type of housing demonstrably influences the separation of paper. We established the null hypothesis as follows: **H0.1.** The type of housing unit of households does not affect the sorting of paper waste.

The results of the statistical analysis show that the paper separation differs depending on the type of housing ($\chi2 = 8.989$, df = 1, $p = 0.003$). The contingency table of standardized residuals indicates a small difference in separation, but greater avoidance of paper separation in single-family homes. We reject the null hypothesis and accept the new hypothesis: **H1.1.** The type of housing unit of households has an impact on the sorting of paper waste.

A total of 176 respondents (11.77%) commented on the reasons for not separating paper, of which 154 respondents (87.51%) selected from four defined reasons and 22 respondents stated another reason (12.49%) (Table 4). The reasons given for not sorting paper waste are different depending on whether the respondents live in IHC or CHC households. The main reason given by IHC respondents is missing containers or paper collection bags (32 respondents, 18.18%), which confirmed hypothesis H2. Common reasons given by IHC and CHC respondents (55 respondents, 31.25%) were: containers are very remote (confirmation of hypothesis H5), containers are often overfilled (25 respondents, 14.21%, confirmation of hypothesis H4), 22 respondents (12.50%) think it is nonsense and believe everything still ends up in the landfill, and 14 respondents admitted that they did not have enough information (7.96%) (confirmation of hypothesis H6). Additional reasons were specified by 22 respondents (12.49%) (Table 4). Of the total 1416 answered respondents (94.72%), 750 respondents (52.97%) chose the option "without reservations about the current conditions of paper collection" and 666 respondents (47.03%) provided us with suggestions for improving paper collection. Of this number, 212 respondents (31.83%, mainly from IHC, where paper collection is carried out with a bag method with a frequency of once a month) suggested replacing the bags used for paper collection with a waste container or increase the number of containers for separates collection at the expense of mixed municipal waste. In the CHC, the respondents expressed their view that it would be beneficial to have denser network of paper containers, or an installation of semi-underground containers of the MOLOK type (117 respondents, 17.57%), or an increase in the frequency of paper collection (118 respondents, 17.72%) Thus, the hypotheses H2, H3 and H4 were confirmed. As many as 47 respondents demanded better information, education and promotion of separation and information images for containers (7.06%) (Table 5), which confirmed hypothesis H5.



**Table 4.** Reasons for not separating paper, plastics, and glass waste in respondents' households in the city of Nitra.

| Reasons for Non-Separation of Paper, Plastics, and Glass | Paper | | Plastics | | Glass | |
|---|---|---|---|---|---|---|
| | Number of Respondents | Percentage (%) | Number of Respondents | Percentage (%) | Number of Respondents | Percentage (%) |
| the paper, plastics, and glass collection container are missing or the containers are very remote | 93 | 52.84 | 72 | 52.55 | 186 | 70.99 |
| nonsense, because all the separated waste still ends up in the landfill | 22 | 12.50 | 20 | 14.60 | 22 | 8.40 |
| the containers are often overcrowded | 25 | 14.21 | 15 | 10.95 | 14 | 5.34 |
| the lack of information on separation of paper, plastics, and glass | 14 | 7.96 | 16 | 11.68 | 15 | 5.73 |
| the paper, plastic, and glass waste is not produced | 2 | 1.14 | 1 | 0.73 | 14 | 5.34 |
| disinterest, comfort, and laziness | 3 | 1.70 | 3 | 2.19 | 4 | 1.53 |
| there are few containers in the area, it is necessary to first increase their volume or frequency of collection | 5 | 2.84 | 3 | 2.19 | 2 | 0.76 |
| absence of motivation to reduce municipal waste | 3 | 1.70 | 3 | 2.19 | 1 | 0.38 |
| absence of space to store paper, plastics, and glass in my household | 4 | 2.27 | 2 | 1.46 | 3 | 1.15 |
| others reasons | 5 | 2.84 | 2 | 1.46 | 1 | 0.38 |
| Total | 176 | 100.00 | 137 | 100.00 | 262 | 100.00 |

**Table 5.** Respondents' suggestions for the current paper, plastics, and glass waste collection in the city of Nitra.

| Suggestions for the Current Collection of Paper, Plastics, and Glass Waste | Paper | | Plastics | | Glass | |
|---|---|---|---|---|---|---|
| | Number of Respondents | Percentage (%) | Number of Respondents | Percentage (%) | Number of Respondents | Percentage (%) |
| there is no collection container for paper, plastic, glass and to provide a container to this waste | 212 | 31.83 | 176 | 32.95 | 228 | 60.16 |
| denser network of containers, increase their capacity or replace them with semi-underground containers | 117 | 17.57 | 104 | 19.47 | 80 | 21.11 |
| increase the frequency of paper, plastics and glass collection | 118 | 17.72 | 141 | 26.40 | 20 | 5.28 |
| better information, education, promotion, pictures with instructions for separation into containers | 47 | 7.06 | 77 | 14.42 | 25 | 6.60 |
| absence of motivation to sort waste, pay only for the amount of waste produced-quantity collection (money-better motivation) | 26 | 3.91 | 10 | 1.88 | 5 | 1.32 |
| failure of the paper, plastics and glass collection process, comments on collection these waste, unclear collection schedule | 57 | 8.55 | 16 | 3.00 | 13 | 3.43 |
| others suggestion | 4 | 0.60 | 1 | 0.19 | 8 | 2.10 |
| no answer or inappropriate answer to the question | 85 | 12.76 | 9 | 1.69 | 0 | 0 |
| Total | 666 | 100.00 | 534 | 100.00 | 379 | 100.00 |

### 3.2.2. Plastics

Overall, 1379 respondents (92.24%) responded to question, whether they separate plastic waste, and the answers were significantly dominated by the answer "yes", which was marked by 1239 respondents (89.85%), 727 respondents were from CHC and 512 respondents were from IHC.

When testing the null hypothesis (**H0.2**: Housing unit type does not affect the classification of plastic waste), we confirmed that the type of housing has no effect on plastic separation ($\chi2 = 1.347$, df = 1, $p = 0.245$). Households from IHC and CHC both have the same approach to the sorting of plastic waste.

The reason for not separating plastic waste was stated by 137 respondents (9.16%). The main reason was that the containers are very remote, or they are missing (72 respondents, 52.55%) (confirmation of hypothesis H2 and H5). Furthermore, 20 respondents think that separation is nonsense, because all waste ends up in landfills (14.60%), 15 respondents cite frequent overfilling of containers (10.95%) (confirmation of hypothesis H4) and 16 respondents admit to the lack of information on separation of plastics (11.68%) (confirmation of hypothesis H5). Another reason was specified by 14 respondents (10.22%) (Table 4). Comments to the separation of plastics chosen 534 respondents (39.21%) (Table 5). In particular, respondents suggest allocating a collection container that is absent, especially in IHC (176 respondents, 32.95%) (confirmation of hypothesis H2) or increase of the frequency of plastic waste collection (141 respondents, 26.40%) (confirmation of hypothesis H4). As with paper waste, plastic waste was collected with bags at the time of completion of the questionnaire at the IHC in the city of Nitra. Other responses included proposals to increase the number of collection containers or increase the volume of plastic collection containers (104 respondents, 19.47%). The answers mainly concerned CHC, which confirmed hypothesis H3. A large part of the population thinks that it would be appropriate to better inform the population what belongs and what does not belong to the plastic collection container, or how to dispose of plastic waste to take up a smaller volume in the collection container (77 respondents, 14.42%) (confirmation of hypothesis H6) (Table 5).

### 3.2.3. Glass

Similarly, as with paper and plastic waste, we were interested in whether the inhabitants also separate glass from other waste. Out of the total number of respondents, 1359 respondents (90.90%) commented. Of the recorded answers, a positive answer was confirmed by 1089 respondents (80.13%), 696 respondents from CHC and 393 respondents from IHC.

Likewise, we were interested as to whether the type of housing demonstrably affects the separation of glass. We established the null hypothesis as follows: **H0.3.** The type of dwelling unit of households does not affect the sorting of glass waste.

The result of the statistical analysis shows that the type of housing demonstrably affects the separation of glass ($\chi2 = 69.727$, df = 1, $p < 0.001$). The contingency table of standardized residues indicates the preference for separating glass in flats (CHC) and avoiding its separation in family houses (IHC). We reject the null hypothesis, and we accept the new hypothesis: **H1.3**: The type of housing unit of households has an effect on the sorting of glass waste.

Total 262 respondents (17.52% of all respondents) were able to state the reasons why they do not separate glass waste; 70.99% specified the missing of containers and their distance from the household (186 respondents) as a reason for not separating glass (confirmation of hypothesis H4); 22 respondents (8.40%) think it is nonsense, because they believe all the separated waste still ends up in the landfill; 15 respondents (5.73%) do not have enough information; and 14 respondents (5.34%) state that the containers are often overfilled. Specific reasons for not separating glass waste were given by 47 respondents from IHC who do not own a container for separated glass in their household. Other reasons were specified by 25 respondents (9.54%) (Table 4).Question about current about glass collection was answered by 1341 respondents (89.70%). Overall, 962 respondents

(71.74%) have no reservations about the current conditions of glass collection in the city of Nitra, but 379 respondents do have reservations (28.26%). As the most common answer, respondents stated that they have neither a container nor a bag for separated glass in the IHC (228 respondents, 60.16%) or that the containers were too remote (confirmation of hypothesis H2 and H5); additional answers stated that the containers for glass are always overfilled (80 respondents, 21.11%), citizens lack sufficient information about what belongs in a glass container and what does not, e.g., in the form of various images on containers (25 respondents, 6.60%), and an increase in the frequency of glass collection would be beneficial (20 respondents, 5.28%), etc. (confirmation of several hypotheses H3, H4 and H6) (Table 5).

### 3.2.4. Bio-Waste, Kitchen Waste and Kitchen Oil Waste

1259 respondents (84.21%) answered the question: "What do you do with bio-waste from the garden, household, or landscaping?" Respondents could choose more than one answer to this question. The survey collected 1758 responses to this question (with 1259 respondents providing answers). Unfortunately, the most common answer was that the bio-waste from the garden and household is mixed into municipal waste (589 respondents, 33.50%) and the waste placed in this way is transported to the landfills with other types of unsorted waste. Similar number of responses (552, 31.40%) showed that this type of waste is placed into brown container (mainly answers of respondents from IHC). According to 335 responses (19.06%) was the bio-waste composted in household composters, vermicomposters or in community composters. Transport of bio-waste to the city's composting plant or the collection yard was selected in 223 responses (12.68%). The bio-waste incineration was found in 22 responses (1.25%). The option "other way" was chosen in 37 responses (2.11%) (Figure 6). Among these answers the most common answer was that the respondents do not have and do not produce this type of waste (20 respondents, 54.06%) or they use it for pets (11 respondents, 29.73%). Other options were represented in answers of respondents in a very low percentage.

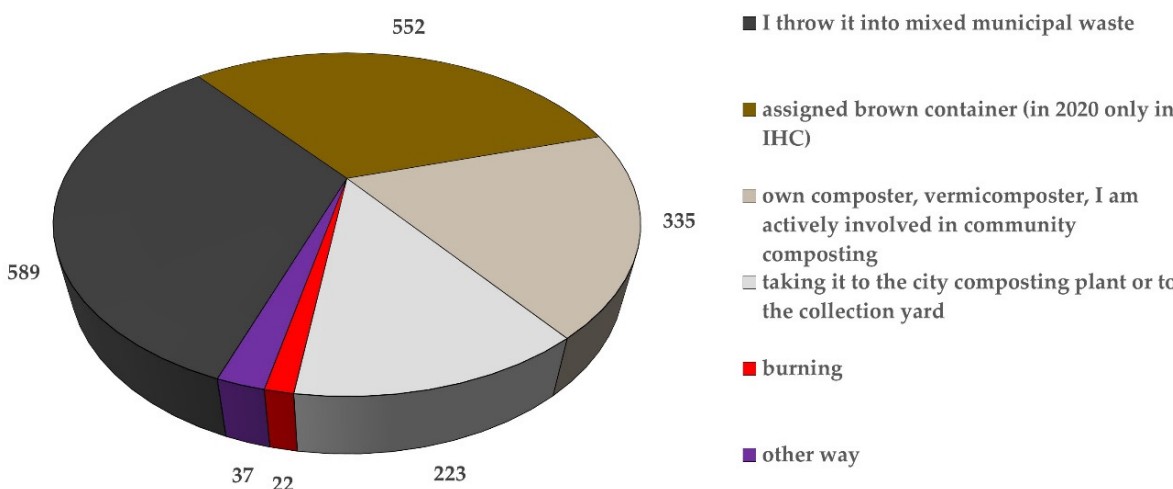

**Figure 6.** Ways of disposing of bio-waste from gardens, households and from the landscaping in responses of the respondents in the city of Nitra.

The results of the questionnaire point to different conditions for bio-waste separation in IHC and CHC. We statistically tested whether the different conditions are demonstrably reflected in the different degree of bio-waste sorting in CHC and IHC. We established the null hypothesis as follows: **H0.4**: Housing unit type does not affect the classification of bio-waste. The type of housing demonstrably influences the separation of bio-waste ($\chi2 = 708.54$, df = 1, $p < 0.001$). We reject the null hypothesis, and we accept the new hypothesis: **H1.4**: The type of housing unit of households has an impact on the sorting of

glass waste. The contingency table of standardized residues shows a high separation of bio-waste in family houses (IHC) and a low separation in apartments (CHC).

As already mentioned, there is a difference between IHC and CHC bio-waste separation conditions. This is confirmed by the responses from IHC. Compared to CHC, the responses showed that respondents have their own composters (347, 23.12%), or they minimize their bio-waste by use as pet food (211, 14.06%) (Figure 7). Only a very small group of the answers of respondents concerned the alternative of a community composting site for CHC, which is built in the city of Nitra (five responses).

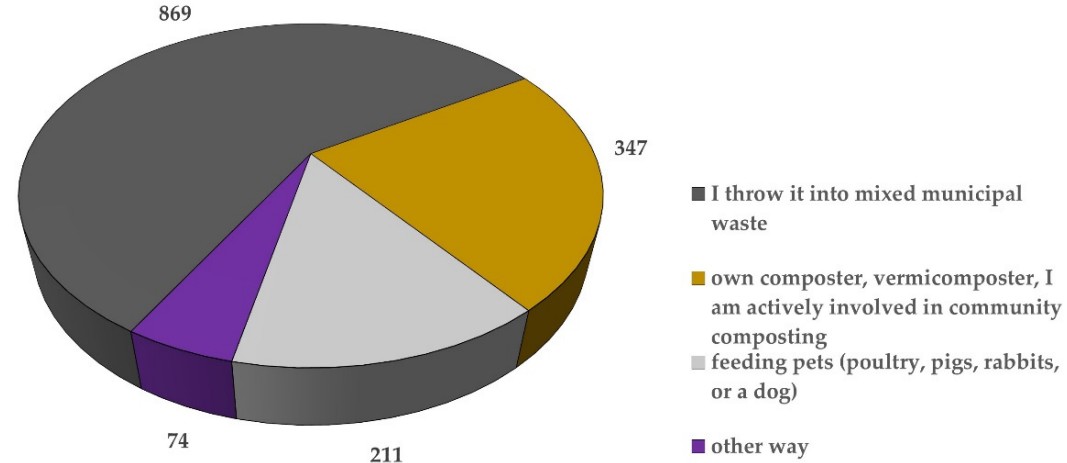

**Figure 7.** Way of handling bio-waste from the kitchen in the households of respondents in the city of Nitra.

The support of composting in the city of Nitra was strengthened by a separate question in the questionnaire. Respondents were able to express whether they would welcome the possibility of the city of Nitra providing citizens with composters for family houses and starting community composting in housing estates. Overall, 1236 respondents (82.68%) answered the question, of which 1066 respondents (86.25%) answered positively. This did not confirm our H7 hypothesis. A total of 180 respondents added further comments specifying their views. The answers were dominated by: "yes, to family homes" (45 respondents, 25.00%), allocation of brown containers to housing estates (CHC) (44 respondents (44, 24.44%), better information of citizens, education, motivation (27, 15.00%), fear of smell, rodents, mess, and unesthetic environment (22, 12.22%) and increase the frequency of bio-waste collection during the season, i.e., from spring to autumn (10, 5.56%). Other options were mentioned only by few respondents (32, 17.78%).

However, it should be noted that a substantial part of the bio-waste produced in the city of Nitra is generated mainly in IHC gardens and in the preparation of public spaces, cemeteries, and greenery (e.g., city park, mowing of grasslands, public greenery in housing estates, city landscaped areas, etc.).

Separately in the questionnaire, we paid attention to kitchen waste, which-in contrast to bio-waste from gardens-is produced in approximately the same amount in both IHC and CHC households. The question about kitchen bio-waste was answered by the same number of respondents as the question about general bio-waste (1259, 84.21%). As before, respondents were able to provide more than one answer to this question. The total number of answers was 1501 (with 1259 respondents providing answers). Comparably to the answers from the previous question, the most common responses included the inappropriate way bio-waste from the kitchen is handled, i.e., disposal to mixed municipal waste and subsequent landfilling (869 respondents, 57.89%). In case of 347 respondents (23.12%), the way of dealing with such waste was composting (composter or vermicomposter) (Figure 7), or alternatively, community composting. Among respondents who stated other ways of handling kitchen bio-waste, 56 from 74 respondents (75.68%) said that such waste is properly handled, and the city of Nitra collects this type of waste from brown containers

(in 2020 only in IHC), and 10 respondents stated that they have a waste shredder and then pour the waste into the toilet or sewer. Other options have answered few respondents.

Also, in the case of kitchen waste, we were interested in whether the type of housing has a significant effect on the separation of kitchen waste. We established the null hypothesis as follows: **H0.5**: The type of dwelling unit of households does not affect the sorting of kitchen waste. The type of housing demonstrably affects the separation of kitchen waste ($\chi^2$ = 183.1, df = 1, *p* <0.001), and for this reason we reject the null hypothesis and accept the new hypothesis: **H1.5**: The type of housing unit of households has an impact on the sorting of kitchen waste. The contingency table of standardized residues shows a high separation of kitchen waste in family houses (IHC) and a low separation in apartments (CHC). The result of this statistical testing was the same as in the case of bio-waste, although households in IHC compost to a greater extent and give leftovers to domestic animals; and at the same time, they are not the dominant creator of kitchen waste.

We separately focused on used kitchen oil waste. A total of 1259 respondents (84.21%) commented on the methods of its disposal. The most common way to dispose of kitchen oil waste in households is by pouring it into the toilet or sink, where it consequently goes into the sewer (292 respondents, 23.19%). A significant proportion of respondents dispose of kitchen oil waste together with mixed municipal waste (211 respondents, 16.76%). Other methods of disposal include handing it over at a petrol station (206 respondents, 16.36%) or at a collection yard (175 respondents, 13.90%) (Table 6).

**Table 6.** Ways of disposing of used kitchen oil waste in the households of respondents in the city of Nitra.

| Way | Number of Answers | Percentage (%) |
|---|---|---|
| I pour into the toilet, or into the sink into the sewer | 292 | 23.19 |
| I throw it into mixed municipal waste | 211 | 16.76 |
| I hand over at the petrol station | 206 | 16.36 |
| I hand over at the collection yard | 175 | 13.90 |
| I don't use kitchen oil; I don't have it | 145 | 11.51 |
| I hand it over at the used-oil containers at shopping malls | 75 | 5.96 |
| by no means, I don't know, I don't care | 71 | 5.64 |
| composting | 28 | 2.22 |
| pets for feeding, into their diet | 20 | 1.59 |
| I use it for burning and heating | 11 | 0.87 |
| others | 25 | 2.00 |
| Total | 1259 | 100.00 |

3.2.5. Other Types of Waste

In the city of Nitra, basic commodities are separated in accordance with applicable legislation. We were interested whether the inhabitants of the city also separate other types of waste. Overall, 1202 respondents (80.40%) answered this question. Respondents chose from several options in this question. The total number of answers we recorded for this question was 3112. Of the options offered in this question and the total number of respondents, most of them said that they separate batteries and accumulators (975 respondents, 81.11%), textiles (772 respondents, 64.23%), electrical waste (745 respondents, 61.98%), fluorescent lamps (486 respondents, 40.43%) and 134 respondents (11.15%) chose the option "other" type of waste, for example metals (18, 13.43%), medicines after expiration (16, 11.94%), small construction debris (12, 8.95%), hazardous waste–paints, varnishes, adhesives, chemicals, solvents, toners (10, 7.46%), iron (8, 5.97%), etc.

The last question regarding waste separation concerned collection yards. We were interested in other waste the respondents handed over to collection yards. Overall, 1251

respondents (83.68%) commented on the question. Respondents were allowed to choose more options for this question. In total, we recorded 3050 answers, which shows that the inhabitants of the city of Nitra use the collection yards for the collection of the following: bulky waste, such as carpets, mattresses, furniture, interior doors, etc. (770 respondents, 61.55% of respondents´ and 25.25% of all answers), small construction debris (632 respondents, 50.52% and 20.72%), electrical waste (559 respondents, 44.68% and 18.33%), metals (403 respondents, 32.21% and 13.21%), hazardous waste (271 respondents, 21.66% and 8.88%) and bio-waste (161 respondents, 12.87% and 5.28%). Collection yards have unsuitable opening hours, they are very remote and therefore not used by 229 respondents (18.31% and 7.51%) and 25 respondents (2.00% and 0.82%) stated "other" than the types of waste listed, which they bring to the collection yard (Table 7), especially glass, cardboard, polystyrene, old clothes, oil, iron, batteries, and accumulators, etc. In general, we can say that if residents use collection yards, it is mainly because the frequency of transport of some commodities (e.g., bio-waste) is insufficient, inhabitants encounter overfilling of containers (e.g., textiles, bulk paper) or there is not container in the vicinity of their residence for the given commodity (e.g., glass).

**Table 7.** Types of waste handed over to collection yards according to respondents in the city of Nitra.

| Type of Waste | Number of Answers | Percentage (%) from the Number of 1251 Respondents | Percentage (%) of the Total Number of All Answers |
|---|---|---|---|
| bulky waste, such as carpets, mattresses, furniture, interior doors, etc. | 770 | 61.55 | 25.25 |
| small construction debris | 632 | 50.52 | 20.72 |
| electrical waste | 559 | 44.68 | 18.33 |
| metals | 403 | 32.21 | 13.21 |
| hazardous waste | 271 | 21.66 | 8.88 |
| I do not use a collection yard | 229 | 18.31 | 7.51 |
| bio-waste (collection from spring to autumn 1 × 2 a week is not enough) | 161 | 12.87 | 5.28 |
| other type of waste | 25 | 2.00 | 0.82 |
| Total | 3050 | 100.00 | 100.00 |

### 3.3. Waste Prevention

Two questions of the questionnaire were focused on waste prevention. Firstly, we asked respondents directly if they were trying to prevent waste; secondly, we were curious about the specific ways in which they were trying to minimize waste. The first question was answered by 1240 respondents (82.94%). The question provided respondents with four options: "Yes, I try to prevent waste"; "No"; "I am not interested in reducing waste"; "I don't know what the question means". Overall, 1112 respondents (89.68%) chose the first option. The answers of the respondents did not confirm our hypothesis H8. A total of 69 respondents (5.56%) chose the second option. The term "waste prevention" is not understood by 30 respondents (2.42%) and 29 respondents (2.34%) are not interested in reducing the amount of waste in their households.

The second question, which focused on specific methods of waste prevention ("zero waste") by the respondents themselves, was answered by 1216 respondents (81.34%). Due to the fact that the respondents could choose more than one answer, the total number of answers was 4213 (1216 respondents). From the provided answers the most common was the answer "I take my own shopping bags or textile bags to the store", which was chosen by 1122 respondents. Of all the respondents (1216), this answer has a high share (92.27%) and also has a high share (26.63%) of the total number of all answers (4213). Other frequently selected options that were repeated by the respondents were "I only buy what

I really need, I am not influenced by advertising" (868 respondents, 71.38% and 20.60%) and "I drink tap water, I do not buy PET bottles" (809 respondents, 66.53% and 19.20%) (Table 8). A total of 72 respondents (5.92% and 1.71%) also chose the "other" option, where the most answers were assigned to the options: non-waste of food, growing in your own garden and the breeding of animals for food purposes (11, 15.28%), shopping clothes in second hands, bazaars, non-shopping by fast fashion (11, 15.28%), minimization of plastic waste (8, 11.11%), repairing a damaged thing or renting, exchanging, sewing, donating, upcycling (8, 11.11%), reusing waste (5, 6.94%),etc.

**Table 8.** Ways of waste prevention according to respondents in the city of Nitra.

| Ways | Number of Answers | Percentage (%) of 1216 Respondents | Percentage (%) of the Total Number of All Answers |
|---|---|---|---|
| I carry my own shopping bags or textile bags to the store | 1122 | 92.27 | 26.63 |
| I only buy what I really need, I am not influenced by advertising | 868 | 71.38 | 20.60 |
| I drink tap water; I don't buy PET bottles | 809 | 66.53 | 19.20 |
| I prefer quality products with a long service life | 520 | 42.76 | 12.34 |
| I buy large "family packages" without unnecessary packaging | 464 | 38.16 | 11.01 |
| I shop in stores without packaging | 206 | 16.94 | 4.90 |
| I shop in drugstores with filling into my own containers | 152 | 12.50 | 3.61 |
| other | 72 | 5.92 | 1.71 |
| Total | 4213 | 100.00 | 100.00 |

### 3.4. Awareness of the Population

We have addressed the two main issues in the area of public awareness with two questions. In the first, we were interested by the means respondents received information about the municipality into their households. There were six options offered: 1. Radničné noviny (Municipal newspaper); 2. City website; 3. City social networks (Facebook page, Instagram page etc.); 4. Other press media (e.g., MY Nitra, Nitrianske ECHO); 5. Discussions, debates, and various events; 6. Free SMS service of the Municipal Office in Nitra. This question was answered by 1212 respondents (81.07%). At the same time, we allowed the respondents to specify other means of information according to their opinions. In this question, the respondents were able to give more answers, so the total number of answers we received was 2453 (from 1212 respondents). The most frequent answer provided by the respondents was that they use the "Facebook page of the city" (824 respondents). This answer has a high share (67.99%) among the number of the respondents (1212) and also has a high share (33.59%) of the total number of all answers provided (2453). The least common answer by the respondents was "Discussions, debates and various events" as a means of information (Table 9). In the "Other" option, most of respondents (32) stated that they do not use the city's information sources, do not know about any of them and have no information about them (61.53%).

**Table 9.** Use of information sources in the city of Nitra according to respondents.

| Way | Number of Answers | Percentage (%) | |
| --- | --- | --- | --- |
| | | of the Number of 1212 Respondents | of the Total Number of All Responses |
| Radničné noviny (Municipal newspaper) | 294 | 24.26 | 11.99 |
| city website | 669 | 55.20 | 27.27 |
| city social networks | 824 | 67.99 | 33.59 |
| other printed media | 441 | 36.39 | 17.98 |
| discussions, debates, and various events | 79 | 6.52 | 3.22 |
| free SMS service of the Municipal Office in Nitra | 94 | 7.76 | 3.83 |
| others | 52 | 4.29 | 2.12 |
| Total | 2453 | 100.00 | 100.00 |

The second question in this part of the questionnaire we wanted to know which information campaign on proper sorting and waste prevention initiated by the city would suit the respondents the most. Overall, 1222 respondents (81.74%) commented on this question. Respondents were able to provide more than one answer, so the total number of answers was 2440 (1222 respondents). From the answers of the respondents, the most common answer was "internet campaign". This method of information campaign on the correct separation and prevention of waste is satisfactory for up to 995 respondents (81.24% of respondents and 40.78% of all answers). The respondents would be least satisfied with the method of the campaign in the form of "debate, event and discussion" (178 respondents, 14.57% and 7.29%) (Table 10). In addition to the 5 options offered, respondents were also able to indicate another option, according to their own views. This option was chosen by 114 respondents (9.33% and 4.67%) and the answers were significantly dominated by the respondents' opinion that the containers or areas near the containers should contain information on the correct separation of waste (30 respondents, 26.31%), next discussions in schools (all types), work with children, presentations in schools (15, 13.16%), leaflets on recyclable paper or a brochure on proper separation (12, 10.52%), social networks (FB, IG) (10, 8.77%), etc. This confirmed our hypothesis H6.

**Table 10.** The form of an information campaign of the city on the correct separation and prevention of waste generation according to the respondents in the city of Nitra.

| Forms | Number of Answers | Percentage (%) | |
| --- | --- | --- | --- |
| | | of the Number of 1222 Respondents | of the Total Number of All Answers |
| internet | 995 | 81.42 | 40.78 |
| local TV | 244 | 19.97 | 10.00 |
| leaflet by mail | 637 | 52.13 | 26.11 |
| Radničné noviny (Municipal newspaper) | 272 | 22.26 | 11.15 |
| debates, events, discussions | 178 | 14.57 | 7.29 |
| others | 114 | 9.33 | 4.67 |
| Total | 2440 | 100.00 | 100.00 |

*3.5. Ideas, Suggestions, and Comments for Improvement*

In the last question of the questionnaire, we left the respondents to express any comments on the current state of municipal waste management in the city of Nitra. We expected the respondents to offer us their ideas, suggestions, and comments to improve of waste management situation. Only 357 respondents (23.88%) answered this question. Of the respondents, 23 respondents did not provide any comments (6.44%). Of all the respondents, up to 109 stated in their suggestions the addition or increase of the number of containers for separate waste collection, including bio-waste collection containers in CHC (30.53%), another 58 respondents (16.25%) suggest increasing the frequency of collection of separated components of municipal waste, due to frequent overfilling of containers and 43 respondents (12.05%) believe that the city should do more in the field of education and information on the proper separation and prevention of waste. The answers of the respondents confirmed several of our hypotheses (H2, H3, H4, and H6). Among the respondents there were also those (38 respondents, 10.65%) who are convinced that everyone should pay only as much for the collection of waste as they produce. This could form a start for the city in terms of motivation towards the citizens. Among the other ideas, suggestions and comments (25, 7.00%) (Table 11), there are also those that represent very good ideas for the city government in the future (e.g., aesthetic modification of the surroundings of container stands, more stands and rubbish bins for dog excrement, SWAP also for gardeners, library of things, e.g., on AX 1 hall "reuse centre", introduction of separate collection (glass, metals, paper, plastics and bio-waste) in cemeteries, introduction of separate collection also in public spaces, littering, etc.

**Table 11.** Ideas, suggestions, and comments of respondents to improve the situation in the field of waste management in the city of Nitra.

| Ideas, Suggestions, and Comments | Number of Answers | Percentage (%) |
| --- | --- | --- |
| addition of containers for bio-waste in CHC and increase the number of containers | 109 | 30.53 |
| more frequent waste collection-proper setting of collection, often overfilled containers | 58 | 16.25 |
| more education and awareness, people lack context, prevent waste | 43 | 12.05 |
| motivation of people-pay relatively to the waste produced (e.g., through the assigned QR code) | 38 | 10.65 |
| no, I don't have any, I don't know about any | 23 | 6.44 |
| community composting of bio-waste in CHC | 16 | 4.48 |
| camera system, control by the city police | 15 | 4.20 |
| control of the population and severe fines for illegally deposited waste | 10 | 2.80 |
| spring, summer, and autumn collective cleaning of the surroundings of your residence | 9 | 2.52 |
| bad setting of collection yard operations (they will not accept all types of waste, unsuitable opening times) | 7 | 1.96 |
| abolition of the fee for small construction debris and asbestos, illegal landfills are then created | 4 | 1.12 |
| others | 25 | 7.00 |
| Total | 357 | 100.00 |

## 4. Discussion

By conducting a questionnaire survey among the citizens of the city of Nitra, the city government obtained significant and valuable information which is important for reducing the volume of managed municipal solid waste. The answers of the respondents confirmed almost all of our predetermined hypotheses (H1–H6).

For the setting up of waste management in the city of Nitra, it is necessary to address the circumstances separately for individual (IHC) and complex housing construction (CHC), as both have their specific starting points and conditions for waste disposal. We also proved this fact by statistical analysis. Our results show that households living in IHC and CHC have different approaches to sorting individual types of waste. While paper and glass sorting are more preferred in CHC, bio-waste and kitchen waste disposal is more preferred in IHC. The reason is that in CHC there are paper and glass containers in apartment buildings, while in IHC these containers are absent. Bio-waste and kitchen waste, however, are managed more actively in households in the IHC, which have the option of disposing of this type of waste by composting or container collection. This method is lacking in CHC. Both types of households (IHC and CHC) approach the separation of plastic waste in the way, which we also proved by statistical analysis. Since we found that respondents are interested in home and community composting (86.25%), hypothesis H7 was not confirmed. We were also surprised by the relatively high percentage of respondents who said that they try to prevent waste generation in several ways (89.68%) and hypothesis H8 was not confirmed.

A Danish study, based on the statistical analysis of 1442 households distributed among 10 individual sub-areas in three Danish municipalities, found that overall waste composition and generation rates were not significantly different between the three municipalities; however, there were difference in the waste composition from single-family and multi-family houses. The waste generation rates were similar for single-family and multi-family houses, although the individual percentage composition of food waste, paper, and glass was significantly different between the housing types [23]. A questionnaire survey in connection with municipal waste management was used in several domestic and foreign studies. Within four different types of Shanghai communities (Shanghai, China), authors identified attitudes and behaviour of citizens not only in a structured questionnaire survey, but also with direct personal interviews [24]. The results indicated that respondents are ecologically aware of separation, but practice only minimal separation. This may be related to a significant proportion of respondents (37.73%) who report a lack of information and awareness about separation. We agree with the authors [25], who state that relatively high levels of information and knowledge do not always necessarily translate into action.

One of the objectives of another paper was to examine and evaluate the main shortcomings and proposed improvements to the collection system according to the respondents [26]. One of the questions asked was: "Do you separate municipal solid waste?" Only 2.3% of the population of Mercato San Severino in southern Italy said that they never separate MSW. In contrast, 81.9%, 10.0% and 5.9% represented the percentage of people who do separate collection "always", "often" and "sometimes" respectively. In questions 4, 7 and 10 in our questionnaire, we tried to identify whether the respondents, i.e., the inhabitants of the city of Nitra, separate paper waste, plastics, and glass. Of all 1495 respondents, 1434 (95.92%) responded to question on paper waste, 1379 (92.24%) on plastics and 1359 (90.90%) on glass. Although the answer "yes" dominated in the answers to all 3 questions, the shares of respondents who do not separate the given commodity are much higher in our results compared to the Italian city of Mercato San Severino (paper 12.55%, plastics 10.15% and glass 19.87%). Of these 3 commodities, the most inhabitants of the city of Nitra are involved in separation of plastics—1239 respondents (89.85%), followed by 1254 respondents (87.45%) who are involved in paper separation and the least—1089 respondents (80.13%)—are involved in glass separation. Similarly, a survey in Shanghai revealed that most people separate only recyclable materials (mainly paper and plastics), and few people separate glass waste.

Differences in the composition and parameters of bio-waste originating from urban settlements (U-bio-waste) and family houses (F-bio-waste) during four seasons, were evaluated by statistical analyzes in the city of Prague (Czech Republic). Larger amounts of bio-waste were produced in family houses (F-bio-waste), which was related to garden activities during the year (especially grass and leaves) [27]. In the case of a study from Shanghai [24], 47.69% of respondents composted biodegradable waste. Unfortunately, the answer in our question 13 was that 589 respondents (33.50%) throw bio-waste from the garden and household incorrectly into the mixed municipal waste. Overall, 552 respondents (31.40%) chose composting as an option. In the SR, the Act of the National Council of the Slovak Republic no. 79/2015 Coll. on waste and on the amendment of certain laws from 2016 [28] introduced not only a ban on the landfilling of biodegradable municipal waste from gardens, parks, and cemeteries (so-called green bio-waste), but also a ban on its incineration (Article 13 Prohibitions, letter g). Nevertheless, in the answers of 22 respondents, we encountered this method of disposal. In this case, too, there is a fine for such a method of disposing of bio-waste (fine for natural persons in the amount of EUR 1500 or EUR 2500 for legal persons, in the case of an offense and not a criminal offense (the damage did not exceed EUR 266).

Overall, 1251 respondents (83.68%) commented on the question 18 whether the inhabitants of the city of Nitra use collection yards and what types of waste they bring to the yards. Respondents were able to provide more than one answer. We also recorded larger number of responses (3050). The most common answers included bulky waste, e.g., carpets, mattresses, furniture, and interior doors (770 respondents, 61.55%). However, it is also worth mentioning the 229 respondents who stated in our questionnaire that they do not use collection yards because they are very remote and have unsuitable opening hours (18.31%). These results do not correspond to the results of a study conducted on a sample of 350 respondents in Ohio, where up to 52% of respondents said they had used one of the waste collection facilities at least once. The average number of visits reported by respondents was 22 times a year, meaning slightly less than every other week. Of those who never used the centre, only 33% knew about the location of the centre in their village. The evaluation of the results showed that the type of dwelling and the distance to the nearest waste collection point did not play a significant role [29]. Insufficient use of collection yards by the inhabitants of the city of Nitra in the long run is probably related to insufficient information of the inhabitants about their importance and function. Collection yards are also used by inhabitants of the city of Nitra to store electrical waste (559 respondents, 44.68%). Compared to the results of the next study [30], the results of the answers of our respondents (year 2019) reached approximately the same percentage of research answers in Poland in 2012 (42%). In Poland, however, in 2014 the collection of electrical waste increased to 68%.

Inhabitants of the city can access the data on the correct separation by various means of information of the city. The results of our questionnaire survey show that the city's social networks (67.99%) and the city's website (55.20%) are the most common. The study in Shanghai showed that television (58.80%) and newspapers (31.71%) represented the two main sources of information on the MSW resources department. The option including Internet and social networks was chosen by 26.39% of respondents. Similar results were obtained from a survey of the recycling behaviour of 456 households in Dhaka (Bangladesh). The main source of information for city residents on recycling is the newspapers (50.20%), followed by other sources (24.90%), television (20.88%) and radio (4.02%) [31]. Although, waste separation at the source is known to be an effective method for sustainable waste management, it assumes that the inhabitants sort the waste correctly. It is not sufficient to merely inform, but it is necessary to educate people about the societal benefits of correct waste management. In order to do this, different communication campaigns need to be developed targeting different groups of consumers [32].

Assessment of the views of stakeholders involved in the separation of municipal solid waste collection program are also presented on the example of interviews with a municipal

manager, 2 recycling workers and 403 citizens in the eastern zone of São Paulo (Brazil). All interviewees highlighted the importance of disseminating environmental education aspects to the population to increase awareness of environmental issues and adopt more effective waste separation practices [33]. Modern society is becoming a–waste society rather than a well-to-do society: the waste that people produce litters our streets and is not always in the bins [26].

In the city of Nitra, we have registered a number of illegal landfills in recent years. In the period from 2017 to 2020, according to official data of the municipal authority in the territory of the city of Nitra, a total of 68 illegal landfills were removed (11 in 2017, 8 in 2018, 27 in 2019 and 22 in 2020). The composition of these landfills varies, they include small construction debris, bio-waste, and plastics. Any person who performs illegal landfilling of waste as a misdemeanour (damage does not exceed EUR 266) according to Article 115 of the Waste Act, can be liable for payment of higher fines (maximum fine for natural persons in the amount of EUR 1500 or EUR 2500 for legal entities). In case of a criminal offense, i.e., the damage exceeds EUR 266, the fine may increase up to EUR 120,000 pursuant to Article 117 (3) of the Waste Act. The use of generously sized waste containers has dramatically reduced the occurrence of illegal dumping [34].

Some respondents expressed an opinion that the payment for small construction debris and asbestos should be abolished. Small construction debris can be handed over by residents only at the collection yard in the premises of the Nitra municipal services, Ltd. and they have to pay a fee in accordance with the General Binding Regulation of the City of Nitra no. 1/2020 [35]–in the amount of EUR 0.05 for 1 kg of such waste. As small construction debris is collected only at 1 collection yard in the city of Nitra, it can discourage residents who have to move this waste from more distant parts of the city. Disposal of asbestos as hazardous waste, or its stabilization before landfilling is performed by specialized experts in accordance with the valid legislation of the SR and EU directives and regulations. Self-help disposal of asbestos-containing materials carries heavy fines (EUR 2500 for natural persons and EUR 2500–120,000 for legal entities).

Directive (EU) 2018/851 of the European Parliament and of the Council on Waste [36] points to examples of incentives for the application of the waste hierarchy, such as landfill and incineration charges and pay-as-you-go schemes, where waste generators pay on the basis of the amount of waste created and which provide incentives for the separation of recyclable waste at source and for the reduction of mixed waste (Annex IVa). This opinion is shared by respondents in our questionnaire survey. We have seen these answers in the following questions: 5 (26 respondents, 3.91%), 11 (5 respondents, 1.32%), 12 (1 respondent, 1.33%), 16 (27 respondents, 15.00%), 22 (2 respondents, 1.75%) and 23 (38 respondents, 10.65%). All the above answers show and represent the opinion that the separation of household waste should be reflected in the amount of the fee for the removal of mixed municipal waste, the so-called quantity collection. Similar views can be found in the answers of respondents from Shanghai, where the method of "payment for the bag" was preferred by 40.30% of respondents and "payment by household size" was preferred by 38.06% of respondents [24]. The city of Mercato San Severino in southern Italy adopted a pay-as-you-throw (PAYT) program in 2005, so citizens are charged for MSW collection based on the amount of waste they throw away [26]. The municipalities often use a flat-rate system with no incentive to reduce the amount of waste produced by the population [37]. Similarly, in the city of Nitra, the same method is used, which can lead to a lack of interest in reducing the amount of waste, as well as less interest in separation.

Municipal waste accounts for approximately 7 to 10% of total waste generated in the EU. However, in terms of its management, it is one of the most demanding waste streams, and the way in which it is managed is usually a good indicator of the quality of the overall waste management system in the country. Problems related to municipal waste management stem from its highly diverse and mixed composition, the location of the generated waste in the immediate vicinity of the population, its considerable visibility in public space and its impact on the environment and human health. As a result, municipal

waste management requires a comprehensive system, including an efficient waste collection system, an efficient system for separation and proper monitoring of waste streams, the active involvement of citizens and businesses, infrastructure adapted to the specific composition of waste and a detailed financing system. Countries that have built effective municipal waste management systems generally perform better in the whole field of waste management, including the achievement of recycling targets. Many EU Member States have not yet fully built up the necessary waste management infrastructure [36]. Slovak Republic (SR) is one of the countries in the EU with the lowest rate of waste separation and recycling. Many types of waste that would still be usable as a secondary raw material end up at landfills unnecessarily. The EU average is currently 45%, while in the SR the recycling rate is around 39%.

We can reduce waste production by changing the consumer behaviour of individuals. As part of the questionnaire survey, we addressed this issue in 2 questions (questions 19 and 20). In question 19, 82.94% of respondents who try to prevent waste generation reacted positively. Among the answers given by the respondents, the following methods were the most common: bringing their own shopping bags to stores, buying only necessary things, drinking tap water, and not buying water in PET bottles, buying large family packages without unnecessary packaging or shopping in non-packaging stores, including bottled drugstore. The issue of how the prevention of generation of waste can affect the total production of household waste is the topic of the next study [38], which compares the amounts of waste produced in 4 different model families living in the SR (without separation, with partial separation, complete zero waste and partial zero waste). While a member in a family without subsequent separation of municipal waste produced 129.54 kg in course of 2019, one member in a family with complete zero waste produced only 2.39 kg in that same period.

The amount of waste produced at the European and global level increases year by year [39]. Over the last 20 years, solid waste has become an important issue in municipalities and counties [34]. Municipal waste is produced by people and have to be managed following legislative, technical, and social rules. A separate collection programme is based on several rules that the citizen has to follow [25]. Waste management is the one service just about every city government provides for its residents. While service levels, environmental impacts and costs vary dramatically, waste management is arguably the most important municipal service and serves as a prerequisite for other municipal activities [1]. Sustainable municipal waste management is regarded as one of the key elements for achieving urban sustainability via mitigating global climate change, recycling resources, and recovering energy [40].

## 5. Conclusions

The growth of waste generation, the high rate of landfilling and the low rate of waste minimization require more efficient waste management than ever before. Reserves in the area of municipal waste management in the city of Nitra are present for both participants in this process, i.e., not only the city local government together with municipal services, but also the residents themselves. According to the answers of the respondents from IHC and CHC in our questionnaire, the main shortcomings of municipal waste management by the city include the following:

-    Insufficient number of collection containers for separated commodities,
-    Improper placement of containers in terms of their distance to the residents,
-    Frequent overfilling of collection containers due to low or insufficient transport frequency,
-    Insufficient information and education of the city's inhabitants in the field of waste separation and minimization,
-    Low quality of information media and missing information boards or panels near collection points for collection containers,

- Missing camera systems in the vicinity of the stands with collection containers in the CHC, to which residents from the surrounding municipalities outside the city of Nitra or residents from the IHC also bring their waste,
- Insufficient control and penalties for infringements,
- Absence of support for community composting in CHC and home composting in IHC,
- Lack of waste separation in public spaces (e.g., stadiums, parks, cemeteries, etc.),
- Improper conditions for the collection yard system (e.g., in terms of opening hours or in terms of the possibility of handing over all types of waste, not only selected ones),
- Low incentives for people, i.e., financial advantage for those who separate and minimize mixed municipal waste.

The results of the questionnaire survey resulted in several suggestions and observations of the respondents, which were included in the process of municipal waste management in the city of Nitra in course of the years 2020–2021:

- Based on the population's interest in composting in the IHC, the city carried out a survey of interest in a domestic composter in two rounds (June, August), and currently the first composters were distributed to IHC,
- From 1 July 2021 the collection of kitchen waste started in housing estates (CHC). For the family houses (IHC) it is planned to start the collection of kitchen waste at the end of 2021,
- When collecting separated waste in the IHC, the bag collection in all parts of the city of Nitra is gradually replaced by 120 L yellow containers for plastic (from 2021),
- In the coming years, the city plans to provide a 120 L collection blue container for paper in parts of the city with IHC,
- In CHC, the city continues with the installation of semi-underground containers, of which the respondents showed great interest in the questionnaire,
- In September 2021, mobile collection of used edible oils and fats in connection with exchange for products such as vinegar or sunflower oil took place,
- A new online circular map of Nitra (https://bit.ly/cirkularna-mapa-nitra; accessed on 25 October 2021) has been made available, where all places supporting zero waste, upcycling and green business can be found,
- In Radničné noviny (https://www.nitra.sk/zobraz/sekciu/radnicne-noviny-10453; accessed on 23 September 2021) and on the website of the city of Nitra (https://www.nitra.sk/; accessed on 2 November 2021), residents are regularly informed about the way, meaning and schedule of classification of individual municipal waste commodities.

Many of the respondents' answers were also used in 2021 in the creation of the prepared strategic document of the city of Nitra for the area of municipal waste management for the years 2022–2027. Unfortunately, due to the pandemic situation of COVID-19, no significant activities have been undertaken in the city of Nitra in the area of education and promotion of the population.

In the process of establishing and improving well-performing municipal waste management systems, the understanding of fundamental social factors to influence household behaviour is commonly underestimated but of utmost importance [41]. The capture rates of recycling vary significantly between fractions of waste, cities, and neighbourhoods, as do the waste separation behaviours of both individuals and households [42]. Obviously, there is not "one applicable system that serves them all", and this is reflected in the differences of municipal waste management system implemented in different areas. The more this system corresponds to local conditions, the more efficient it is [41]. As participation of individual household members in separating waste fractions is crucial for an effective recycling process, understanding motivations for people to take part in municipal recycling systems is important [42]. The principal aspect in the area of the sustainable consumption is the effective education of consumers, as well as implementation of positive changes in their awareness, behaviours, trends, and attitudes. They should also be open to changes related to the implementation of pro-environmental solutions in everyday life because, without the conscious involvement of consumers, it is impossible to achieve success in the

area of sustainable consumption [39] and hence, at all layers of the waste management hierarchy [40].

In conclusion, we can state that in accordance with the main goal of this paper and based on the information obtained from the respondents' answers, not only the municipality of Nitra, but especially the residents themselves, have a long and though journey ahead in the process of municipal waste management.

**Author Contributions:** Conceptualization and methodology—Z.P., I.J., A.B. and J.K.; data preparation— Z.P., I.J., A.B. and J.K.; data analysis—Z.P.; data curation—Z.P., I.J. and A.B.; writing—Z.P. and I.J.; visualization—Z.P. and I.J., supervision—Z.P.; correspondence—I.J. All authors have read and agreed to the published version of the manuscript.

**Funding:** This research was supported by the project No. 052/2020/OPII/VA "Scientific support of climate change adaptation in agriculture and mitigation of soil degradation" (implemented within the ERDF, Operational Programme Integrated Infrastructure).

**Institutional Review Board Statement:** Not applicable.

**Informed Consent Statement:** Not applicable.

**Data Availability Statement:** Not applicable.

**Conflicts of Interest:** No potential conflict of interest was reported by the authors.

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
