# Peer review of "Streamlining the Municipal Waste Management System in the City of Nitra (Slovak Republic) Based on a Public Survey"

_sustainability, doi:10.3390/su132413992_

Round 1
Reviewer 1 Report
This reviewer finds some merits in this paper. For example, the method implemented is quite comprehensive with a well-structured framework shown in Figure 2. There are lots of findings from this case study. However, the paper is not structured properly to ease the understanding of the key findings.
Some of the key drawbacks are highlighted below:
- Title of the article: The title is too long for a research paper. Authors need to revise the title to better reflect its content.
- Abstract appears as an executive summary of a report. It is lengthy, not written succinctly. Line numbers 18-22 may not be necessary in an abstract. Authors need to summarize the findings concisely.
- Proper structuring and paragraphing of the manuscript sections need to be done. For example, in the Introduction section, some paragraphs are lengthy (3rdand 4th) and some are just one sentence (2nd). Authors need to follow one message per paragraph norm for the readers to follow the points easily.
- There are 28 tables in this manuscript. They are just presented without proper synthesis of the key findings. Without synthesizing the key findings, it merely looks like a report. It is very difficult for an independent reader to grasp the key points from this research.
- Section 4 – Discussion and Conclusion – to be completely restructured. Need to be arranged in sub-titles with key messages.
- The key findings of the study should be separately summarized as Conclusions
Author Response
Author's Reply to the Review Report (Reviewer 1)
We would like to thank to Reviewer 1 for his/her comments and suggestions for manuscript improvement. We tried to correct the manuscript according to them.
This reviewer finds some merits in this paper. For example, the method implemented is quite comprehensive with a well-structured framework shown in Figure 2. There are lots of findings from this case study. However, the paper is not structured properly to ease the understanding of the key findings.
Some of the key drawbacks are highlighted below:
Point 1
Title of the article: The title is too long for a research paper. Authors need to revise the title to better reflect its content.
Answer 1
We've shortened and edited the title of the article. From the original " Improving the efficiency of municipal waste management in the city of Nitra, Slovakia by using the attitudes and opinions of its inhabitants (report from the questionnaire survey)" to the new " Streamlining the municipal waste management system in the city of Nitra (Slovak Republic) using the results of the questionnaire".
Point 2
Abstract appears as an executive summary of a report. It is lengthy, not written succinctly. Line numbers 18-22 may not be necessary in an abstract. Authors need to summarize the findings concisely.
Answer 2
We have shortened the abstract, deleted unnecessary parts and supplemented the summary findings based on the recommendations.
Point 3
Proper structuring and paragraphing of the manuscript sections need to be done. For example, in the Introduction section, some paragraphs are lengthy (3rdand 4th) and some are just one sentence (2nd). Authors need to follow one message per paragraph norm for the readers to follow the points easily.
Answer 3
We changed the structure of the manuscript, unified the paragraphs and deleted unnecessary irrelevant parts of the results.
Point 4
There are 28 tables in this manuscript. They are just presented without proper synthesis of the key findings. Without synthesizing the key findings, it merely looks like a report. It is very difficult for an independent reader to grasp the key points from this research.
Answer 4
We have joined tables for individual types of waste. We have added statistical analyzes, syntheses and better linked text with tables. As part of the synthesis of key findings, we independently interpreted the degree of separation in individual housing construction and complex housing construction.
Point 5
Section 4 – Discussion and Conclusion – to be completely restructured. Need to be arranged in sub-titles with key messages. The key findings of the study should be separately summarized as Conclusions
Answer 6
We have separated the discussion from the conclusions. We added several syntheses to the discussion and included the key findings of the study in the conclusions.
Reviewer 2 Report
The manuscript with the title “Improving the efficiency of municipal waste management in the city of Nitra, Slovakia by using the attitudes and opinions of its inhabitants (report from the questionnaire survey)” contains interesting information, however it looks more like a technical report than a research article. In my opinion the methodological approach must be improved and a statistical analysis of the results carried out. Furthermore, the analysis of the literature should be deepened.
General comments:
Introduction:
- The state of the art about the topic needs a clear improvement: literature about previous sociological surveys aimed at citizens on waste management is completely missing
- The goal of this study should be made clearer
Materials and Methods:
- It is not clear whether the respondents’ sample is representative of the population. I suggest the authors provide the characterization of sample and discuss the issues of using a convenience sample.
- The statistical analysis of the results is completely missing; the authors should do so
- Many information on waste contained in paragraph 2.1 should be included in the supplementary material as well as tables 1-3
Results:
- The Results section is too long; the main aspects should be summarised highlighting only the most important results. All other aspects can be put in the supplementary material.
- The questionnaire results should be elaborated by applying a statistical analysis in order to explore whether the intrinsic characteristics of the respondents influence their opinions and behaviours about waste management.
Discussion and conclusions:
- The discussion section is too poor, I suggest the authors to enrich it with more considerations and a broader comparison with the literature.
- I think it would be preferable to put the conclusions in a separate section.

Author Response
Author's Reply to the Review Report (Reviewer 2)
We would like to thank to Reviewer 2 for his/her comments and suggestions for manuscript improvement. We tried to correct the manuscript according to them.
The manuscript with the title “Improving the efficiency of municipal waste management in the city of Nitra, Slovakia by using the attitudes and opinions of its inhabitants (report from the questionnaire survey)” contains interesting information, however it looks more like a technical report than a research article. In my opinion the methodological approach must be improved and a statistical analysis of the results carried out. Furthermore, the analysis of the literature should be deepened.
General comments:
Introduction:
Point 1
The state of the art about the topic needs a clear improvement: literature about previous sociological surveys aimed at citizens on waste management is completely missing
Answer 1
Unfortunately, in the city of Nitra, previous sociological surveys focused on citizens and waste management have not been solved so far. We have already pointed out this fact in the modified version of the article.
Point 2
The goal of this study should be made clearer
Answer 2
We have completely reformulated the goal based on the recommendation. We have defined the new goal as follows: “The aim of our contribution was to obtain, analyze and interpret the attitudes and opinions of the inhabitants of the city of Nitra, through an anonymous questionnaire, and use them to streamline the municipal waste management system in the city. “
Materials and Methods:
Point 3
It is not clear whether the respondents’ sample is representative of the population. I suggest the authors provide the characterization of sample and discuss the issues of using a convenience sample.
Answer 3
We have added a more detailed description of the sample of respondents to the manuscript (lines 268-274).
Point 4
The statistical analysis of the results is completely missing; the authors should do so
Answer 4
We supplemented the results with a statistical comparison of the degree of separation in individual housing construction and complex housing construction. This helped us to interpret the results better as well as to make recommendations for practice.
Point 5
Many information on waste contained in paragraph 2.1 should be included in the supplementary material as well as tables 1-3 The number of tables actually made the text of the article very disruptive.
Answer 5
We combined the tables for individual types of waste into larger tables. We have cleaned the other tables from unnecessary irrelevant parts.
Results:
Point 6
The Results section is too long; the main aspects should be summarised highlighting only the most important results. All other aspects can be put in the supplementary material.
The questionnaire results should be elaborated by applying a statistical analysis in order to explore whether the intrinsic characteristics of the respondents influence their opinions and behaviours about waste management.
Answer 6
We deleted all unnecessary parts from the results, unified the tables and changed the structure of the results. At the same time, we added statistical evaluation as well as their interpretations to the results.
Discussion and conclusions:
Point 7
The discussion section is too poor, I suggest the authors to enrich it with more considerations and a broader comparison with the literature.
I think it would be preferable to put the conclusions in a separate section.
Answer 7
We have separated the discussion from the conclusions. We added several syntheses to the discussion and included the key findings of the study in the conclusions.
Reviewer 3 Report
Improving the Efficiency of Municipal Waste Management in the City of Nitra, Slovakia, by using the attitudes and opinions of its inhabitants (report from the questionnaire survey)
The article discusses the opinions and attitudes of citizens in Nitra city, Slovakia, regarding waste management, which was gathered through survey. The questionaries consist of some questions about residences’ attitudes in managing their own various waste types and any solutions that they might have regarding solid waste management. In general, the study provides valuable insights in such topics and can answer the question as to why Slovakia has one of the lowest rate of waste segregation and recycling in the European Union, subject to incorporation/addressal of the following technical and general comments:
Technical Comments
- Was the questionnaire intended to be filled by each household or residents in Nitra city nor any other specific despondences targeted to fill the questionaries. Please explain.
- How can the authors provide solutions regarding diaper waste management in Slovakia, which was said to be one of the critical environmental problems throughout the globe?
- How can the authors use the answers and suggestions from the questionnaire to advise a future waste management plan for the Slovakia, especially in the years of 2021-2035?
General Comments
- Authors need to shortened the abstract.
- Please use the same term for mentioning the country of Slovakia, whether it is Slovakia or Slovak Republic.
- Lines 94-95: specify the meaning of ‘gradual implementation of various measures.
- Line 98: although it has been explained that SR stands for Slovak Republic in the previous section, it is better to rewrite SR as Slovak Republic (SR) as line 98 is a different section.
- Lines 163-203: the current problems of waste management in the Slovak Republic is suggested to be moved to the Introduction section of the paper.
Author Response
Author's Reply to the Review Report (Reviewer 3)
We would like to thank to Reviewer 3 for his/her comments and suggestions for manuscript improvement. We tried to correct the manuscript according to them.
The article discusses the opinions and attitudes of citizens in Nitra city, Slovakia, regarding waste management, which was gathered through survey. The questionaries consist of some questions about residences’ attitudes in managing their own various waste types and any solutions that they might have regarding solid waste management. In general, the study provides valuable insights in such topics and can answer the question as to why Slovakia has one of the lowest rate of waste segregation and recycling in the European Union, subject to incorporation/addressal of the following technical and general comments:
Technical Comments
Point 1
Was the questionnaire intended to be filled by each household or residents in Nitra city nor any other specific despondences targeted to fill the questionaries. Please explain.
Answer 1
We have added a more detailed description of the sample of respondents to the manuscript (lines 268-274)
Point 2
How can the authors provide solutions regarding diaper waste management in Slovakia, which was said to be one of the critical environmental problems throughout the globe?
Answer 2
With a second question in the questionnaire and a short paragraph in the article about disposable baby diapers, we wanted to emphasize this issue, which is also addressed in research articles. At the same time, we wanted to point out the amount of waste from disposable baby diapers as unnecessarily generated waste, if in case reusable diapers would be used.
Point 3
How can the authors use the answers and suggestions from the questionnaire to advise a future waste management plan for the Slovakia, especially in the years of 2021-2035?
Answer 3
The results of the questionnaire are very valuable information for the city government. The municipality is able to set the management of municipal waste so that suitable conditions are provided for the inhabitants in the area of waste separation, which will contribute to the city of Nitra in meeting the set goals in the field of waste management for the coming years.
General Comments
Point 4
Authors need to shortened the abstract.
Please use the same term for mentioning the country of Slovakia, whether it is Slovakia or Slovak Republic. Lines 94-95: specify the meaning of ‘gradual implementation of various measures. Line 98: although it has been explained that SR stands for Slovak Republic in the previous section, it is better to rewrite SR as Slovak Republic (SR) as line 98 is a different section. Lines 163-203: the current problems of waste management in the Slovak Republic is suggested to be moved to the Introduction section of the paper.
Answer 4
We have shortened the abstract, deleted unnecessary parts and supplemented the summary findings. We have also incorporated all the other general comments of the reviewer into the current version of the manuscript
Round 2
Reviewer 1 Report
- The revised title of the paper seems much better. However, the reviewer is of the opinion that the text given in bracket '(report from the questionnaire survey)' can be removed
- The Abstract has been revised. But, it should be further shortened to bring the number of words within the set limit of the journal.
- The quality of the tables (For example: Tables 5, 6, 7, 8, 12 and 13) can be improved with proper formatting.
Author Response
Response to the Reviewer 1 comments:
We would like to thank to Reviewer 1 for his/her comments and suggestions for next manuscript improvement. We tried to correct the manuscript according to them.
Point 1
The revised title of the paper seems much better. However, the reviewer is of the opinion that the text given in bracket '(report from the questionnaire survey)' can be removed
Answer 1
We removed the "report from the questionnaire survey" section from the title of the paper, and based on the advice of the Academic Editor we modified the title of the paper as follows: "Streamlining the municipal waste management system in the city of Nitra (Slovak Republic) based on a public survey".
Point 2
The Abstract has been revised. But, it should be further shortened to bring the number of words within the set limit of the journal.
Answer 2
We modified the part of the abstract, shortened it and removed parts that are not essential. The final abstract has 247 words.
Point 3
The quality of the tables (For example: Tables 5, 6, 7, 8, 12 and 13) can be improved with proper formatting.
Answer 3
Based on the Instructions for Authors, we adjusted formatting of the paper, including tables. According to recommendations of Reviewer 2, we replaced some of the tables with a graph. Number of tables was reduced (see tracking changes in the corrected manuscript).
Reviewer 2 Report
The reviewer thanks the Authors for the revised text and the answers to the comments.
The manuscript has been improved a lot.
However, in my opinion, one aspect can be further improved: all quantitative data and results were presented in tables. I suggest to the authors, where possible, to use graphs to show the results thus eliminating some tables. This would make the manuscript more attractive to readers while solving the problem of the excessive number of tables.

Author Response
Response to the Reviewer 2 comments:
We would like to thank to Reviewer 2 for his/her comments and suggestions for next manuscript improvement. We appreciate the comment of the Reviewer for his/her observation of improvements performed in the corrected manuscript.
Point 1
The reviewer thanks the Authors for the revised text and the answers to the comments. The manuscript has been improved a lot.
However, in my opinion, one aspect can be further improved: all quantitative data and results were presented in tables. I suggest to the authors, where possible, to use graphs to show the results thus eliminating some tables. This would make the manuscript more attractive to readers while solving the problem of the excessive number of tables.
Answer 1
Thanks also for this recommendation. In the case of some tables, we really realized that the graph has a better informative value. In some cases, however, the created graph had no informative value, so we kept the original tables. In total, we replaced 5 tables in the text with graphic outputs. Consequently, we renumbered the tables and graphs in the corrected manuscript (see tracking changes).